# Artificial Intelligence Can Guide Antibiotic Choice in Recurrent UTIs and Become an Important Aid to Improve Antimicrobial Stewardship

**DOI:** 10.3390/antibiotics12020375

**Published:** 2023-02-11

**Authors:** Tommaso Cai, Umberto Anceschi, Francesco Prata, Lucia Collini, Anna Brugnolli, Serena Migno, Michele Rizzo, Giovanni Liguori, Luca Gallelli, Florian M. E. Wagenlehner, Truls E. Bjerklund Johansen, Luca Montanari, Alessandro Palmieri, Carlo Tascini

**Affiliations:** 1Department of Urology, Santa Chiara Regional Hospital, 38123 Trento, Italy; 2Institute of Clinical Medicine, University of Oslo, 0315 Oslo, Norway; 3IRCCS “Regina Elena” National Cancer Institute, 00144 Rome, Italy; 4Department of Urology, Campus Bio-Medico University of Rome, 00128 Rome, Italy; 5Department of Microbiology, Santa Chiara Regional Hospital, 38123 Trento, Italy; 6Centre of Higher Education for Health Sciences, 38122 Trento, Italy; 7Department of Gynecology and Obstetrics, Santa Chiara Regional Hospital, 38123 Trento, Italy; 8Department of Urology, University of Trieste, 34127 Trieste, Italy; 9Department of Health Science, School of Medicine, University of Catanzaro, 88100 Catanzaro, Italy; 10Clinic for Urology, Pediatric Urology and Andrology, Justus Liebig University, 35390 Giessen, Germany; 11Department of Urology, Oslo University Hospital, 0315 Oslo, Norway; 12Institute of Clinical Medicine, University of Aarhus, 8000 Aarhus, Denmark; 13Department of Medicine (DAME), Infectious Diseases Clinic, University of Udine, 33100 Udine, Italy; 14Department of Urology, University of Naples Federico II, 80138 Naples, Italy

**Keywords:** urinary tract infection, recurrence, artificial intelligence, antibiotic resistance

## Abstract

Background: A correct approach to recurrent urinary tract infections (rUTIs) is an important pillar of antimicrobial stewardship. We aim to define an Artificial Neural Network (ANN) for predicting the clinical efficacy of the empiric antimicrobial treatment in women with rUTIs. Methods: We extracted clinical and microbiological data from 1043 women. We trained an ANN on 725 patients and validated it on 318. Results: The ANN showed a sensitivity of 87.8% and specificity of 97.3% in predicting the clinical efficacy of empirical therapy. The previous use of fluoroquinolones (HR = 4.23; *p* = 0.008) and cephalosporins (HR = 2.81; *p* = 0.003) as well as the presence of *Escherichia coli* with resistance against cotrimoxazole (HR = 3.54; *p* = 0.001) have been identified as the most important variables affecting the ANN output decision predicting the fluoroquinolones-based therapy failure. A previous isolation of *Escherichia coli* with resistance against fosfomycin (HR = 2.67; *p* = 0.001) and amoxicillin-clavulanic acid (HR = 1.94; *p* = 0.001) seems to be the most influential variable affecting the output decision predicting the cephalosporins- and cotrimoxazole-based therapy failure. The previously mentioned *Escherichia coli* with resistance against cotrimoxazole (HR = 2.35; *p* < 0.001) and amoxicillin-clavulanic acid (HR = 3.41; *p* = 0.007) seems to be the most influential variable affecting the output decision predicting the fosfomycin-based therapy failure. Conclusions: ANNs seem to be an interesting tool to guide the antimicrobial choice in the management of rUTIs at the point of care.

## 1. Introduction

Antibiotic resistance is one of the biggest public health challenges for clinicians and health care systems worldwide, with high costs and a high risk of death due to antibiotic-resistant infection [1,2]. Antibiotic resistance is a particular challenge in urology, due to the high prevalence of bacterial infections among urological patients, and because many urological conditions increase the risk of infections [3,4,5]. In this sense, several authors have demonstrated that ascertaining a correct approach to the management of symptomatic recurrences is a key step towards improving adherence to antimicrobial stewardship principles, due to the rate of rUTIs’ treatment failure (about 35%) [6]. The management of UTIs and the prescription of antibiotics in urology should be performed in accordance with international guidelines’ recommendations and the accurate evaluation of all risk factors impacting on the risk of antimicrobial resistance [7,8]. During the last decade, we have witnessed raised interest in tools able to process large datasets and perform cognitive tasks that facilitate clinical decision-making in urology [7]. In recent years, artificial intelligence (AI) has been progressively applied in different medical specialties, improving either performance of major clinical diagnostic procedures or adherence to recommended treatments [9]. Several authors have demonstrated the efficacy of AI in predicting clinical outcomes in oncological as well as non-oncological diseases [7,10,11,12]. An Artificial Neural Network (ANN) is a machine learning model that allows a computer system to learn from clinical experiences and predict an event from raw data or independent variables. The ANN is based on connections between nodes (the so-called artificial neurons) analogous to brain synapses. ANNs may identify non-linear relationship variables, representing the ideal tool for bridging the gap between linear statistical models and human factors. In brief, ANNs are based on a collection of connected units called “artificial neurons”; each connection, as well as the synapses in a biological brain model, can transmit signals to other neurons. The signal transmission is controlled by the interconnections between the artificial neurons. Artificial neurons may have a threshold such that a signal is sent from one to the other only if the aggregated signal crosses that threshold [10]. In this scenario, we hypothesized that ANNs could be applied as a guide for antimicrobial choice in patients with recurrent UTIs (rUTIs). The aim of this study is to evaluate a neural network made to predict the clinical efficacy of the empirical antibiotic treatment of symptomatic UTI episodes in a large cohort of women affected by rUTIs.

## 2. Results

Of 1234 patients, 1043 patients were finally selected and enrolled in this study, while 191 were excluded for missing data. The patients’ mean age was 39.6 years (SD ± 7.1). All patients’ demographics, clinical and microbiological data at baseline are described in Table 1.

### 2.1. Clinical and Microbiological Data

The mean number of episodes per year was 5.1 (SD ± 1.2). The mean rate of treatment failure was 42% in the 3 month period before enrollment. No differences in terms of anamnestic, clinical or microbiological data have been reported between the two groups (Table 1). Among 1043 patients included in the analysis, 1912 strains were isolated. The most common isolated strains at the baseline evaluation were *Escherichia coli* (1202; 62.9%), followed by *Klebsiella* spp. (374; 19.6%) and *Enterococcus* spp. (199; 10.4%). All microbiological data are displayed in Table 1.

### 2.2. Antibiotic Susceptibility Profiles and Antibiotic Resistance Patterns

The overall mean resistance rate among *Escherichia coli* to ciprofloxacin was 34.7% (418/1202), to cephalexin 8.9% (108/1202) and to fosfomycin 6.6% (80/1202). Finally, 513 out of the 1912 (26.8%) strains were considered as cefpodoxime-resistant and belonged to the ‘ESBL subgroup’. All data about antibiotic susceptibility profiles are detailed in Table 2.

### 2.3. Univariate and Multivariate Analyses

Multivariate analysis identified the following parameters as the most important variables affecting the output decision in predicting fluoroquinolones-based therapy failure: previous use of fluoroquinolones (yes, HR = 4.23, *p* = 0.008) and cephalosporins (yes, HR = 2.81, *p* = 0.003) in the last three months, and the presence of *Escherichia coli* with resistance against cotrimoxazole (yes, HR = 3.54, *p* = 0.001). Finally, the previously mentioned *Escherichia coli* with resistance against cotrimoxazole (yes, HR = 2.35, *p* < 0.001) and amoxicillin-clavulanic acid (yes, HR = 3.41, *p* = 0.007) was the most influential variable affecting the output decision in predicting the fosfomycin-based therapy failure. The previous use of fosfomycin in the last 3 months was not identified as a variable affecting the output decision in predicting fosfomycin therapy failure. No significant differences are reported stratifying data in terms of isolated strains. All univariate and multivariate analysis results are displayed in Table 3.

### 2.4. Artificial Neural Network Analysis

After the completion of the ANN learning and prediction processes, our neural network showed a sensitivity of 87.8% and specificity of 97.3% in predicting the clinical efficacy of empirical therapy. The area under the ROC curve was 0.867 (Figure 1). The calibration plot of predicted probabilities compared with actual probabilities is displayed in Figure 2. ANN analysis reported a high correlation between expected and observed data (train phase: Pearson’s coefficient, 81.5—Root Mean Square error, 1.3; test phase: Pearson’s coefficient, 79.7—Root Mean Square error, 1.2). Moreover, the ANN analysis shows a prediction with an overall accuracy of 83.63%, and the previous use of fluoroquinolones and cephalosporins in the last three months as well as the presence of *E. coli* with resistance against cotrimoxazole to be the most influential variables affecting the fluoroquinolone-based therapy failure. The presence of *Escherichia coli* with resistance to fosfomycin and amoxicillin-clavulanic acid has not been confirmed to be an influential variable affecting the output decision in predicting the cephalosporins and cotrimoxazole-based therapy failure in the ANN analysis. Finally, the previously mentioned *Escherichia coli* with resistance against cotrimoxazole (yes, HR = 2.35, *p* < 0.001) and resistance against amoxicillin-clavulanic acid (yes, HR = 3.41, *p* = 0.007) have been identified as the most important variables affecting the ANN output decision predicting the fosfomycin-based therapy failure.

## 3. Discussion

Here, we demonstrated the feasibility and reliability of ANN applications to guide antimicrobial choice in the empiric treatment of uncomplicated cystitis in everyday clinical practice. The ANN’s performance was good when predicting recurrence in patients with rUTIs. We showed that the previous use of a specific class of antibiotic is not a risk factor for developing bacterial resistance to the same class (except for the fluoroquinolones), but instead the most important risk factor for predicting resistance is the use of other classes of antibiotics. This is a new approach to the treatment of rUTIs.

### 3.1. Results in the Light of Current Knowledge: Nomogram and ANN

Proper data classification and critical management analysis represent an exponentially growing, unmet need in the area of rUTIs. Advances in pharmacological treatment, microbiological pathways and increased antibiotic resistance have contributed to the generation of Big Data in this complex scenario. Due to this increased availability of representative variables, urologists continuously face new challenges when synthesizing information in order to develop accurate diagnosis, tailored treatment plans and outcome prediction in patients affected by rUTIs [5]. Nevertheless, the development of computer-based prediction and decision support models has boosted improvements in the management of rUTIs across different healthcare profiles [13]. Current examples are nomograms or calculators that have been introduced in the clinical management of patients with rUTIs [13,14,15,16]. Recently, Ozkan et al. demonstrated that a model based on an ANN is also a successful medical decision-support system [13] for UTIs with complex symptoms, and the use of this tool is associated with a lower diagnosis cost and shorter diagnosis time [13]. In this context, our study showed promising findings. By using ANN software, the AI showed a sensitivity of 87.8% and specificity of 97.3% in predicting clinical and microbiological efficacy of prescribed antimicrobial medication at a 1 month follow-up.

### 3.2. Results in the Light of Current Knowledge: Previous Antibiotic Exposure and Resistances

Following Cox regression analysis, previous use of fluoroquinolones or cephalosporins were the only significant variables affecting the output decision predicting the fluoroquinolones-based therapy failure in the last three-month period of the identification of *Escherichia coli* strain resistance to cotrimoxazole. Moreover, the presence of a previous *Escherichia coli* infection resistant to either fosfomycin or amoxicillin-clavulanic acid in at least one antibiogram were both indicators of cephalosporins- and cotrimoxazole-based therapy failure. Based on Cox regression analysis, we demonstrated that among all standard clinical and microbiological variables, only the previous use of different antibiotic types was an independent predictor of clinical failure. These data are comparable with the results of a previously published series. Evident disparities between linear statistical or mathematical analysis may justify results obtained in our model, as either rUTIs or a progression to more severe infectious diseases (e.g., pyelonephritis or sepsis) may be related to a complex network of anatomical or clinical variables not available in our dataset. Despite this, the ANN showed the ability to analyze a non-linear relationship, by using multiple layers of interconnected nodes with an adequate accuracy. According to our series, previous microbiological data results seem to have a significant prognostic role in the determination of antibiotic selection. Among rUTIs, only patients prone to multiple resistances to a single *Escherichia coli* strain could benefit from a follow-up with strict observations. The optimal follow-up schedule of rUTIs should not be planned only on the basis of clinical and microbiological variables, rather it should consider the complex interactions between multiple features included in our system. We previously believed that only a small number of rUTIs which reported multiple previous antibiotic treatments should undergo a restrictive follow-up with frequent and more invasive instrumental diagnostic methodologies. However, future studies with a closer follow-up control to check microbiological eradication should be planned to test this hypothesis. The decision to include only adult women in this study is based on a higher incidence of rUTIs in this specific population. Considering their intrinsic anatomical predisposition, adult women must be treated properly to avoid recurrences, severe complications and unnecessary morbidity. Moreover, as highlighted by Cai et al., clinical management shows poor adherence to current antibiotic stewardship, with a subsequent high rate of overdiagnosis and overtreatment [5]. We argue that the use of an ANN enhances antibiotic stewardship adherence in follow-up decision making for this population of patients. Recently, Schinkel et al. developed a machine learning model for predicting blood culture outcomes highlighting the usefulness of learning machines for improving antimicrobial stewardship in a hospital setting [17]. Finally, Yelin et al. demonstrated that the combination of clinical and microbiological parameters by using a learning machine reduced the risk of mismatched treatment when compared with the current standard of care [18]. However, several legal issues should be considered and discussed in the future [19]. In analogy to previous reports, our findings encourage the use of mathematical applications to improve the management of rUTIs in everyday clinical practice [11,20]. We demonstrated the feasibility and reliability of ANN applications in rUTIs, reporting good accuracy in antibiotic selection and hence an improved antibiotic stewardship. If our findings are confirmed in future studies, ANNs should be regarded as a valuable tool to improve the management of patients with uncomplicated UTIs, prevent antimicrobial resistance and improve antimicrobial stewardship. Finally, the findings of this research provide an easy-to-use and low-cost tool for improving our adherence to the principles of antibiotic stewardship in everyday clinical practice. The costs are associated with the software use and the ANN application, as reported by other authors [21].

### 3.3. Strengths and Limitations of This Study

The ANN represented a quick methodology to support the management of patients with rUTIs by computing a combination of established prognostic factors, e.g., previous use of any antibiotics in the last three months and urinary culture results that are used in everyday urological practice. Although it is supported by a large cohort of women, this study has some limitations. Firstly, the characteristics of the patient population. We enrolled only women affected by a recurrent and uncomplicated UTI. This cohort of patients is, however, very common in everyday clinical practice and these patients undergo several antimicrobial treatments and have a high risk of recurrence. We believe this feature explains the prevalence of ESBL strains, which is 26.8%. The suboptimal follow-up timeframe in this specific patients’ cohort remains to be resolved in future studies.

## 4. Materials and Methods

### 4.1. Study Design, Population and Data Source

We conducted a retrospective study including 1043 consecutive patients who had undergone antimicrobial treatment for uncomplicated cystitis between January 2012 and December 2020, at two referral urological centers. Each patient underwent antimicrobial therapy in accordance with the recommendations of EAU guidelines on urological infections [7]. All data were prospectively collected in a dedicated database. All data about previous use of antibiotics and antibiograms were included in the analysis together with clinical follow-up data. We analyzed the importance of bacterial strain, susceptibility and previous antibiotic use for clinical efficacy. Data were analyzed by using the commercially available software program NeuralWorks (NeuralWare, version 3.2, ©Solvusoft Corporation, Carnegie, PA, USA, 2011–2022), as reported by Cai et al. [11]. The results of the ANN were compared with the follow-up data in order to test the feasibility and reliability of AI applications to guide antimicrobial choice in patients with recurrent cystitis.

### 4.2. Inclusion and Exclusion Criteria

We considered all women >18 years affected by clinically and microbiologically confirmed recurrent cystitis for the enrolment. Recurrent cystitis was defined as three episodes of a UTI in the previous 12 months, or two episodes in the previous six months [7]. Only women who tested positive for uropathogens in two or more consecutive cultures (colony-forming units ≥10^5^/mL) were included and analyzed. In line with Cai et al., the following characteristics were considered as exclusion criteria: major concomitant diseases, upper urinary tract congenital abnormalities or stones and sexually transmitted infections [22].

### 4.3. Microbiological Considerations, Follow-Up Evaluation and Outcome

All microbiological and laboratory analyses were performed as previously described [22,23,24]. In particular, the cut-off for significant pathogenic growth was considered ≥10^5^ CFU/mL for the microbiological diagnosis of UTIs [22,24]. Fosfomycin susceptibilities were determined using both the CLSI agar dilution and disk diffusion reference methods as well as using an E-test, in line with Karlowsky J.A. [25] and Tutone M. [26]. Moreover, in line with Tutone et al., isolates resistant to cefpodoxime were inferred to be ESBL-producers [26]. Here, we considered data of the first follow-up (T1). In line with Cai et al., all patients underwent a clinical assessment and microbiological analyses at baseline (T0) and during a follow-up at 1 month (T1) [22]. At the follow-up evaluation, all women underwent urological and midstream urine culture evaluation. The outcome of interest was clinical cure, defined as symptom relief and return to a pre-UTI performance status. Treatment failure was defined as evidence of microbiological colonization with at least 10^5^ Colony Forming Units per milliliter (CFUs/mL) of a single uropathogen in the midstream urine culture associated with the persistence of lower urinary tract symptoms [22,27,28].

### 4.4. Statistical Analysis and Artificial Neural Networks

To evaluate the correlation between the different parameters we used Pearson’s coefficient and Fisher’s exact test to assess the statistical significance with *p* < 0.05 accepted as significant. Moreover, we used Bonferroni’s correction as well as an ANOVA test for a univariate analysis and the log-rank test (Mantel Cox) for a multivariate analysis. We used the Cox proportional hazard regression model to calculate the hazard ratio (HR) with the 95% confidence interval (95% CI). The best cut-off value, in the ANN analysis, was tested by the receiver operating characteristic curve (ROC). The efficacy of the ANN in guiding the antimicrobial choice in the management of recurrent cystitis was tested with accuracy, sensitivity, specificity, positive predictive value and negative predictive value. The error between corresponding pairs of values in two series of values has been evaluated by the Root Mean Square. The ANN methodology and data analysis were described by Cai et al. [11]. All statistical analyses were performed using SPSS 22 for Apple-Macintosh (SPSS, Inc. Chicago, IL, USA). In the present study we evaluated the impact of previous use of antibiotics (all antibiotic classes) in the last three months and the previous resistance (all antibiotic classes) extracted by the last microbiological analysis (antibiogram). All factors were finally used as input parameters (input neurons) to the ANN. All characteristics of the ANN structure were the same as described by Cai et al. [11]. Of 1043 selected patients, 725 (69.5%) were used to train the system, while 318 (30.5%) were used for the testing phase. We did not perform the re-training of the network after the first training phase. No standard logistic regression was performed to compare these results to those of the ANN analysis, because the ANN analysis produces more accurate results than the LR and other standard statistical analyses [29].

### 4.5. Ethical Considerations

This study was planned with a retrospective design; for this reason, Ethical Committee approval was not required in accordance with Italian law. However, all anamnestic, clinical and laboratory data containing sensitive information about patients were de-identified to ensure the analysis of anonymous data only. The de-identification process was performed by non-medical staff by means of dedicated software [30]. The study was conducted in line with the Good Clinical Practice guidelines and the ethical principles laid down in the latest version of the Declaration of Helsinki.

## 5. Conclusions

We demonstrated that an ANN is a feasible and reliable instrument to guide antimicrobial choice in the empiric treatment of uncomplicated cystitis in clinical urological practice, and to predict the recurrence of a UTI. Compared to traditional linear statistical approaches, the ANN provided, in terms of decision-making and prediction accuracy, a significant benefit in terms of adherence to antibiotic stewardship principles in the management of rUTIs in adult women. The ANN enables the clinician to consider more variables of importance for antimicrobial stewardship and has the potential to become an important aid in the management of rUTIs. The isolation of *Escherichia coli* with resistance against cotrimoxazole is the most influential variable for predicting the fluoroquinolones-based therapy failure, and the resistance against cotrimoxazole and amoxicillin-clavulanic acid predicts fosfomycin-based therapy failure.

## Figures and Tables

**Figure 1 antibiotics-12-00375-f001:**
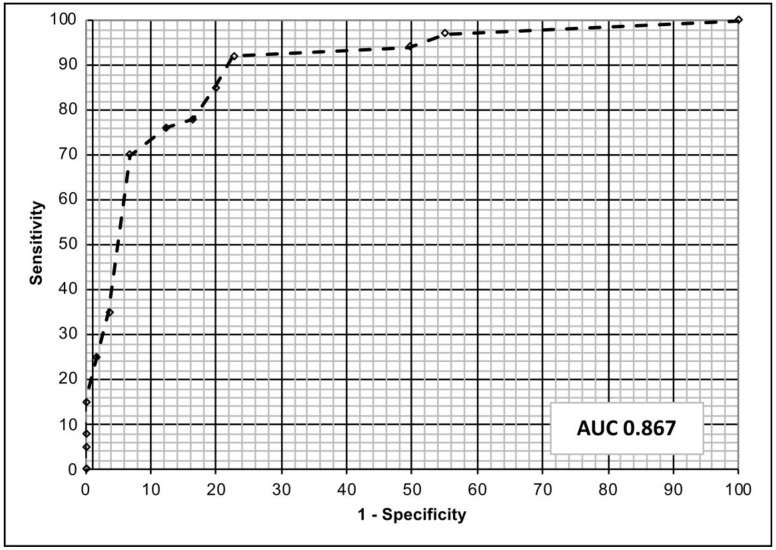
The figure shows the discriminatory performance of our model for predicting the outcome of empirical antibiotic treatment of uncomplicated cystitis. The area under the receiver operating curve (ROC) was 0.867.

**Figure 2 antibiotics-12-00375-f002:**
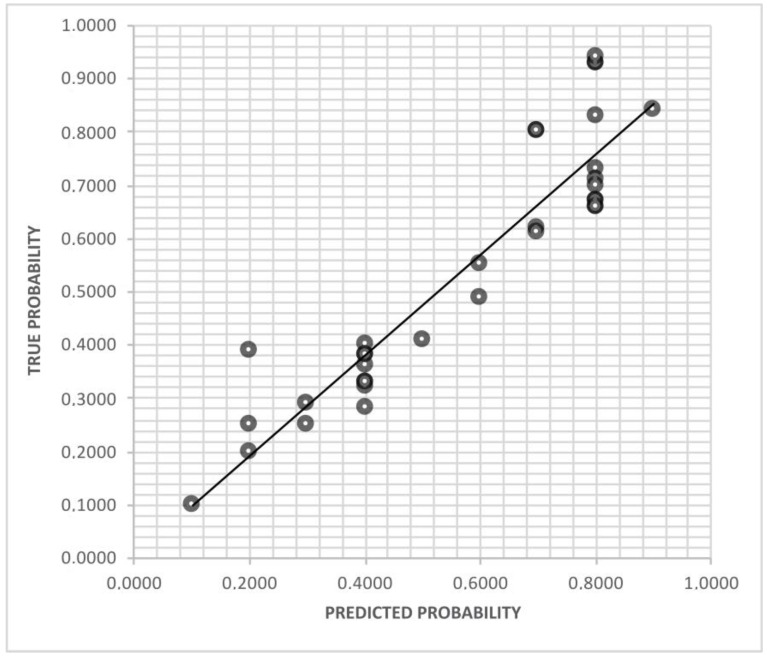
The figure shows the calibration plot of predicted probabilities compared with actual probabilities.

**Table 1 antibiotics-12-00375-t001:** Patient anamnestic, clinical and microbiological characteristics.

				*p*
	Overall	Train Phase	Testing Phase	
No. of analyzed and enrolled patients	1043	725	318	
Age				0.83
Mean (±SD ^†^)	39.6 (±7.1)	39.7 (±7.2)	39.4 (±6.9)	
Marital status				0.31
Married	490 (46.9)	333 (45.9)	157 (49.3)	
Single	553 (53.1)	392 (54.1)	161 (50.7)	
Sexual intercourse per week				1.0
Mean (±SD ^†^)	1.6 (±0.4)	1.6 (±0.5)	1.6 (±0.2)	
Hormonal status				0.75
Premenopausal	991 (95.0)	688 (94.9)	303 (95.2)	
Postmenopausal	52 (5.0)	37 (5.1)	15 (4.7)	
Parity				0.94
Nulliparity	365 (34.9)	253 (34.8)	112 (35.3)	
Multiparity	678 (65.1)	472 (65.2)	206 (64.7)	
Number of UTIs ^#^/year				0.21
Mean (± SD ^†^)	3.1 ± 1.2	3.0 ± 1.1	3.1 ± 1.4	
Previous use of antibiotics (in the last 3 months)				1.0
Yes	448 (42.9)	311 (42.8)	137 (43.0)	
No	595 (57.1)	414 (57.2)	181 (57.0)	
Previous treatment of ABU ^§^				0.83
Yes	564 (54.1)	391 (53.9)	173 (54.4)	
No	479 (43.9)	334 (46.1)	145 (45.6)	
Bacterial strains (1912 isolated from 1043 patients)		*1330 strains*	*582 strains*	
*E. coli*	1202 (62.8)	842 (63.4)	360 (62.1)	
*Klebsiella* spp.	374 (19.6)	259 (19.5)	115 (19.7)	
*Enterococcus* spp.	199 (10.5)	134 (10.0)	65 (11.2)	
*Enterococcus faecalis*	177 (88.9)	123 (91.7)	54 (83.1)	
*Enterococcus faecium*	22 (11.1)	11 (8.3)	11 (16.9)	
*Proteus mirabilis*	52 (2.7)	35 (2.6)	17 (2.9)	
*Enterobacter* spp.	44 (2.3)	31 (2.4)	13 (2.1)	
*Pseudomonas* spp.	41 (2.1)	29 (2.1)	12 (2.0)	

The table shows all patient anamnestic, clinical and microbiological characteristics. SD ^†^ = Standard Deviation; ABU ^§^ = Asymptomatic bacteriuria; UTIs ^#^: urinary tract infections.

**Table 2 antibiotics-12-00375-t002:** Antibiotic resistance of the most common isolated strains to the tested antibiotics.

No. of Isolated Strains	1912					
	*Escherichia* *coli*	*Klebsiella* spp.	*Enterococcus* spp.	*Proteus* *mirabilis*	*Enterobacter* spp.	*Pseudomonas* spp.
Number of isolated strains	1202	374	199	52	44	41
Number of resistant strains and resistance rate (%)						
Amikacin	228 (18.9)	60 (16.1)	24 (12.0)	5 (9.6)	1 (3)	11 (26.8)
Amoxicillin	803 (66.8)	125 (33.5)	65 (32.6)	7 (13.4)	-	-
Cefalotin	781 (64.9)	202 (54.0)	199 (100)	7 (13.4)	-	-
Cephalexin	108 (8.9)	37 (9.8)	199 (100)	6 (11.5)	5 (11.3)	-
Ceftriaxone	348 (28.9)	192 (51.3)	199 (100)	4 (7.6)	6 (13.6)	-
Ciprofloxacin	418 (34.7)	201 (53.7)	96 (48.2)	11 (21.1)	1(2.2)	22 (53.6)
Fosfomycin	80 (6.6)	77 (20.5)	44 (11.7)	4 (7.6)	2 (4.5)	2 (4.8)
Gentamicin	132 (10.9)	115 (30.7)	87 (43.7)	13 (25)	3 (6.8)	16 (39.0)
Levofloxacin	379 (31.5)	207 (55.3)	92 (46.2)	6 (11.5)	1 (2.2)	-
Meropenem	-	-	-	-	-	-
Nitrofurantoin	105 (8.7)	84 (22.4)	25 (12.5)	-	36 (81.2)	-
Piperacillin/Tazobactam	132 (10.9)	120 (32.1)	34 (17.0)	3 (5.7)	3 (6.8)	8 (19.5)
Cotrimoxazole	184 (15.3)	62 (16.5)	33 (16.5)	11 (21.1)	3 (6.8)	-
ESBL (cefpodoxime R)	339 (28.2)	171 (45.7)	-	1 (1.9)	-	2 (4.8)

Table 2 shows the antibiotic resistance of the most common isolated strains to the tested antibiotics.

**Table 3 antibiotics-12-00375-t003:** Univariate and multivariate analysis results of laboratory and clinical factors affecting the output decision in predicting treatment outcome and treatment failure.

Categories (Variables)	Univariate Analysis (p) HR * (95% CI ^†^)	Multivariate Analysis (p) HR * (95% CI ^†^)
Fluoroquinolones therapy failure		
Previous use of antibiotics (last 3 months)		
Aminoglycosides	0.55 (HR 1.05; 95% 0.48–1.61)	0.06 (HR 1.33; 95% 0.75–1.45)
Aminopenicillins	0.06 (HR 1.24; 95% 0.79–1.63)	0.07 (HR 1.12; 95% 0.73–1.35)
Fluoroquinolones	0.03 (HR 3.61; 95% 3.88–4.53)	0.008 (HR 4.23; 95% 3.88–4.53)
Fosfomycin	0.58 (HR 1.32; 95% 1.19–1.67)	0.08 (HR 1.11; 95% 0.97–1.69)
Cephalosporins	0.01 (HR 1.61; 95% 0.87–1.79	0.003 (HR 2.81; 95% 1.95–3.45)
Nitrofurantoin	0.09 (HR 0.70; 95% 0.55–1.00)	0.09 (HR 1.01; 95% 0.70–1.27)
Cotrimoxazole	0.27 (HR 0.71; 95% 0.10–1.12)	0.12 (HR 0.80; 95% 0.45–1.03)
Previous isolated strain		
*Escherichia coli*	0.001 (HR 3.01; 95% 2.94–4.56)	0.002 (HR 2.99; 95% 1.98–3.40)
*Klebsiella* spp.	0.38 (HR 0.92; 95% 0.18–1.27)	0.21 (HR 1.01; 95% 0.97–1.38)
*Enterococcus* spp.	0.21 (HR 1.21; 95% 0.65–1.32)	0.33 (HR 1.82; 95% 0.94–1.30)
*Proteus* spp.	0.08 (HR 1.01; 95% 0.92–1.21)	0.07 (HR 1.18; 95% 0.80–1.49)
*Enterobacter* spp.	0.06 (HR 1.14; 95% 0.70–1.60)	0.09 (HR 1.21; 95% 0.99–1.84)
*Pseudomonas* spp.	0.45 (HR 1.05; 95% 0.80–1.69)	0.71 (HR 1.73; 95% 0.89–1.89)
Previous isolation of *E. coli* resistant to antibiotics		
Aminoglycosides	0.18 (HR 0.97; 95% 0.45–1.11)	0.34 (HR 0.37; 95% 0.28–0.79)
Aminopenicillins	0.10 (HR 1.00; 95% 0.74–1.07)	0.13 (HR 1.28; 95% 0.77–1.34)
Fluoroquinolones	-	-
Fosfomycin	0.87 (HR 1.21; 95% 0.72–1.24)	0.69 (HR 1.14; 95% 0.34–1.04)
Cephalosporins	0.91 (HR 1.78; 95% 0.21–1.15)	0.62 (HR 1.50; 95% 0.77–1.48)
Nitrofurantoin	0.55 (HR 1.03; 95% 0.74–1.91)	0.78 (HR 1.73; 95% 0.91–1.93)
Cotrimoxazole	0.001 (HR 2.99; 95% 2.65–3.19)	0.001 (HR 3.54; 95% 2.73–5.52)
Cotrimoxazole and cephalosporins therapy failure		
Previous isolation of *E. coli* resistant to antibiotics		
Aminoglycosides	0.25 (HR 1.25; 95% 0.68–1.43)	0.31 (HR 1.26; 95% 0.92–1.83)
Aminopenicillins	0.04 (HR 1.97; 95% 1.11–2.02)	0.001 (HR 1.94; 95% 1.20–2.89)
Fluoroquinolones	0.22 (HR 1.17; 95% 0.65–1.23)	0.53 (HR 1.43; 95% 0.77–1.83)
Fosfomycin	0.03 (HR 2.34; 95% 2.03–2.97)	0.001 (HR 2.67; 95% 2.15–3.23)
Cephalosporins	-	-
Nitrofurantoin	0.54 (HR 1.09; 95% 0.82–1.04)	0.91 (HR 1.08; 95% 0.73–1.17)
Cotrimoxazole	-	-
Fosfomycin therapy failure		
Previous isolation of *E. coli* resistant to antibiotics		
Aminoglycosides	0.12 (HR 0.29; 95% 0.10–1.03)	0.39 (HR 0.64; 95% 0.22–0.89)
Aminopenicillins	0.03 (HR 2.53; 95% 2.74–4.01)	0.007 (HR 3.41; 95% 2.27–4.04)
Fluoroquinolones	0.71 (HR 1.90; 95% 0.53–2.13)	0.89 (HR 1.25; 95% 0.89–1.77)
Fosfomycin	-	-
Cephalosporins	0.12 (HR 1.04; 95% 0.56–1.39)	0.28 (HR 0.85; 95% 0.32–1.00)
Nitrofurantoin	0.56 (HR 1.99; 95% 0.79–2.91)	0.10 (HR 1.54; 95% 0.51–1.80)
Cotrimoxazole	0.003 (HR 1.94; 95% 1.27–2.38)	0.001 (HR 2.35; 95% 1.89–3.18)

Table 3 shows the univariate and multivariate analysis results of laboratory and clinical factors affecting the output decision in predicting treatment outcome and treatment failure. **HR** * = Hazard Risk; **CI**
^†^ = Confidence Interval.

## Data Availability

Due to ethical restrictions, clinical data are not available.

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
