# Peer review of "Artificial Intelligence Can Guide Antibiotic Choice in Recurrent UTIs and Become an Important Aid to Improve Antimicrobial Stewardship"

_antibiotics, 2023, doi:10.3390/antibiotics12020375_

Round 1

Reviewer 1 Report

Interesting, but some points need to be improved:

- In the introduction section discuss more about the Artificial Neural Network (ANN) machine. 

- At Lines 167-169 authors reported " Furthermore, results obtained from ANN were used to identify the most significant variables affecting the output decision in predicting clinical response of rUTIs in adult women" What do authors mean? improve please.

- At Lines 203-206 "Finally, Yelin et al. demonstrated that the combination of clinical and microbiological parameters by using a learning machine reduced the risk of mismatched treat ment as compared with current standard of care" Compression algorithm is an essential part of Telemedicine systems, these systems can be correlated with some legal issues. Look at these refs: -- PMID: 28558318 DOI: 10.1016/j.compbiomed.2017.05.024   --- PMCID: PMC9210220 DOI: 10.1177/21925682221090891  - DOI: 10.3389/fnins.2016.00604

- In the discussion section can authors report the cost analysis of thisArtificial Neural Networks (ANN) ? PMCID: PMC9697730  -

- "We demonstrated that ANN is a feasible and reliable instrument to guide antimicrobial 296 choice in the empiric treatment of uncomplicated cystitis in clinical urological practice, 297 and to predict recurrence of UTI." In conclusion section. Add more.

Author Response

Interesting, but some points need to be improved:

- In the introduction section discuss more about the Artificial Neural Network (ANN) machine.

  1. In line with your suggestion, the following sentence has been added to the introduction: “In brief, ANNs are based on a collection of connected units called “artificial neurons”; each connection, as well as the synapses in a biological brain model, can transmit a signal to other neurons. The signal transmission is controlled by the interconnections between the “artificial neurons”. Artificial neurons may have a threshold such that a signal is sent to the other only if the aggregate signal crosses that threshold [9]”.

- At Lines 167-169 authors reported " Furthermore, results obtained from ANN were used to identify the most significant variables affecting the output decision in predicting clinical response of rUTIs in adult women" What do authors mean? improve please.

  1. In line with your suggestion, the following sentence has been deleted: “Furthermore, results obtained from ANN were used to identify the most significant variables affecting the output decision in predicting clinical response of rUTIs in adult women”.

- At Lines 203-206 "Finally, Yelin et al. demonstrated that the combination of clinical and microbiological parameters by using a learning machine reduced the risk of mismatched treatment as compared with current standard of care" Compression algorithm is an essential part of Telemedicine systems, these systems can be correlated with some legal issues. Look at these refs: -- PMID: 28558318 DOI: 10.1016/j.compbiomed.2017.05.024   --- PMCID: PMC9210220 DOI: 10.1177/21925682221090891  - DOI: 10.3389/fnins.2016.00604

  1. In line with your suggestion, the following sentence has been added to the discussion, as well as the following reference: “However, several legal issues should be considered and discussed in the future [27].”
  2. Hejrati, B.; Fathi, A.; Abdali-Mohammadi, F. A new near-lossless EEG compression method using ANN-based reconstruction technique. Comput Biol Med 2017, 87, 87-94.

- In the discussion section can authors report the cost analysis of thisArtificial Neural Networks (ANN) ? PMCID: PMC9697730  -

  1. Many thanks for your suggestion. The following sentence has been added to the discussion: “Finally, the findings of this research provide an easy to use and low-cost tool for improving our adherence to the principles of antibiotic stewardship in everyday clinical practice. The costs are associated to the software use and the ANN application, as reported by other authors [28].” The following sentence has been added too: 28. Aznan, A.; Gonzalez Viejo, C.; Pang, A.; Fuentes, S. Rapid Detection of Fraudulent Rice Using Low-Cost Digital Sensing Devices and Machine Learning. Sensors (Basel) 2022, 22(22), 8655.

- "We demonstrated that ANN is a feasible and reliable instrument to guide antimicrobial 296 choice in the empiric treatment of uncomplicated cystitis in clinical urological practice, 297 and to predict recurrence of UTI." In conclusion section. Add more.

  1. Many thanks for your suggestion.

Reviewer 2 Report

The author submitted a manuscript entitled "Artificial intelligence can guide antibiotic choice in recurrent UTI and become an important aid to improve antimicrobial stewardship", which is an interesting proposal using AI, investigated the feasibility of using ANN to predict and improve the treatment of recurrent UTI.

The authors raise some interesting issues.

I have a few minor observations:

-In Table 1, clarify what ABU means.

- Please review the entire manuscript and tables, bacterial names should be italicized.

- Although this is a retrospective study using previous data, it is missing how "BLEE production" was determined.

Author Response

The author submitted a manuscript entitled "Artificial intelligence can guide antibiotic choice in recurrent UTI and become an important aid to improve antimicrobial stewardship", which is an interesting proposal using AI, investigated the feasibility of using ANN to predict and improve the treatment of recurrent UTI.

The authors raise some interesting issues.

I have a few minor observations:

-In Table 1, clarify what ABU means.

  1. Many thanks for your comment. The Table 1 legend has been revised.

- Please review the entire manuscript and tables, bacterial names should be italicized.

  1. Many thanks for your comment. The manuscript has been revised according with your suggestion.

- Although this is a retrospective study using previous data, it is missing how "BLEE production" was determined.

  1. Isolates that were resistant to cefpodoxime were inferred to be ESBL-producers (Tutone et al.). In this sense, the following sentence has been added to the materials and methods section: “Moreover, in line with Tutone et al, isolates resistant to cefpodoxime were inferred to be ESBL-producers [18]”.

Reviewer 3 Report

The manuscript by Cai et al provides an interesting and useful analysis of the potential for machine learning to improve effective use of antibiotics for treatment of urinary tract infections (UTIs). The work is admirable, and the manuscript is quite publishable, but I do have some suggestions for improvement and readability. 

  1. In the introduction, the authors should provide perspective from the literature on estimates of the rate of UTI treatment failure to put this study in context, given that the goal of the work is to improve the clinical efficacy of antimicrobial treatment of UTIs.  

  2. As best I can tell, the authors do not actually indicate the rate of treatment failure (as defined on line 259) for the patients and UTI events analyzed in their data set. They report the mean number of UTIs per patient in the time period covered by the data set, and one could assume that many of those infections effectively resulted from treatment failures, but perhaps not all of them. This data should be included, as it also provides a baseline for understanding the potential for an ANN with an overall accuracy of prediction of ~ 84% to actually decrease the rate of treatment failure in an analogous patient population. 

  3. Line 88 - More clarity about how the authors defined the “ESBL subgroup” would be useful. If it’s simply based on cefpodoxime resistance, why isn’t that antibiotic simply included in the list in Table 2?  

  4. On line 148 (first paragraph of Discussion section), the authors state that “previous use of a specific class of antibiotic is not a risk factor for developing bacterial resistance to the same class…” This doesn’t appear to be accurate, as Table 3 clearly shows that previous use of fluoroquinolones is a significant risk factor for subsequent fluoroquinolone treatment failure.

  5. Following that, line 149-150 could be revised to state that “the use of other classes of antibiotics is an important factor for predicting the development of ESBL resistance.” However, I don’t think this is what the authors were actually trying to say, as there is no data to support it. They may have been intending to note that previous use of a cephalosporin was a predictor of resistance to another class of antibiotic (fluoroquinolones), as indicated in Table 3. This is somewhat alluded to on lines 124 and 176, and in my opinion this is one of the more interesting and unexpected findings here, as it suggests a connection between those types of resistance. (This is also consistent with, for example, the abundance of ST131 E. coli, which tend to be resistant to both classes of antibiotics, as major contemporary causes of UTIs.) 

  6. Readability for the manuscript would be improved by breaking both the introduction and the discussion (specifically section 3.1) into multiple topical paragraphs, rather than one continuous paragraph.

  7. Reference 4 doesn’t appear to actually show up in the text of the manuscript.

Author Response

The manuscript by Cai et al provides an interesting and useful analysis of the potential for machine learning to improve effective use of antibiotics for treatment of urinary tract infections (UTIs). The work is admirable, and the manuscript is quite publishable, but I do have some suggestions for improvement and readability.

In the introduction, the authors should provide perspective from the literature on estimates of the rate of UTI treatment failure to put this study in context, given that the goal of the work is to improve the clinical efficacy of antimicrobial treatment of UTIs.

  1. In line with your suggestion, the sentence “In this sense, several authors demonstrated that a correct approach to the management of symptomatic recurrences is a key step to improve adherence to antimicrobial stewardship principles [5-6]” has been replaced by the following: “In this sense, several authors demonstrated that a correct approach to the management of symptomatic recurrences is a key step to improve adherence to antimicrobial stewardship principles, due to the rate of rUTI treatment failure (about 35%) [5-6]”.

As best I can tell, the authors do not actually indicate the rate of treatment failure (as defined on line 259) for the patients and UTI events analyzed in their data set. They report the mean number of UTIs per patient in the time period covered by the data set, and one could assume that many of those infections effectively resulted from treatment failures, but perhaps not all of them. This data should be included, as it also provides a baseline for understanding the potential for an ANN with an overall accuracy of prediction of ~ 84% to actually decrease the rate of treatment failure in an analogous patient population.

  1. Many thanks for your comment. The following sentence has been included in the results section: “The mean rate of treatment failure was 42% in the previous 3 months period before enrolment”.

Line 88 - More clarity about how the authors defined the “ESBL subgroup” would be useful. If it’s simply based on cefpodoxime resistance, why isn’t that antibiotic simply included in the list in Table 2? 

  1. Isolates that were resistant to cefpodoxime were inferred to be ESBL-producers (Tutone et al.). In this sense, the following sentence has been added to the materials and methods section: “Moreover, in line with Tutone et al, isolates resistant to cefpodoxime were inferred to be ESBL-producers [18]”. Cefpodxime R were added in table 2.

On line 148 (first paragraph of Discussion section), the authors state that “previous use of a specific class of antibiotic is not a risk factor for developing bacterial resistance to the same class…” This doesn’t appear to be accurate, as Table 3 clearly shows that previous use of fluoroquinolones is a significant risk factor for subsequent fluoroquinolone treatment failure.

  1. The phrase was changed accordingly to your suggestion: previous use of a specific class of antibiotic is not a risk factor for developing bacterial resistance to the same class (except for the fluoroquinolones)…”.

Following that, line 149-150 could be revised to state that “the use of other classes of antibiotics is an important factor for predicting the development of ESBL resistance.” However, I don’t think this is what the authors were actually trying to say, as there is no data to support it. They may have been intending to note that previous use of a cephalosporin was a predictor of resistance to another class of antibiotic (fluoroquinolones), as indicated in Table 3. This is somewhat alluded to on lines 124 and 176, and in my opinion this is one of the more interesting and unexpected findings here, as it suggests a connection between those types of resistance. (This is also consistent with, for example, the abundance of ST131 E. coli, which tend to be resistant to both classes of antibiotics, as major contemporary causes of UTIs.)

  1. Many thanks for your suggestion. The sentence was changed accordingly: We could show that previous use of a specific class of antibiotic is not a risk factor for developing bacterial resistance to the same class (except for the fluoroquinolones), but the most important risk factor for predicting resistance is the use of other classes of antibiotics. This is a new approach to the treatment of rUTI.

Readability for the manuscript would be improved by breaking both the introduction and the discussion (specifically section 3.1) into multiple topical paragraphs, rather than one continuous paragraph.

  1. Many thanks for your comment.

Reference 4 doesn’t appear to actually show up in the text of the manuscript.

  1. Many thanks for your suggestion.

Round 2

Reviewer 2 Report

All the suggestions were correctly addressed.